

# A benzochalcone derivative synchronously induces apoptosis and ferroptosis in pancreatic cancer cells

Xiaoqing Guan[1,2], Bing Zhao[1], Xiaodan Guan[1], Jinyun Dong[1] and Jieer Ying[1,2]

[1] Zhejiang Cancer Hospital, Hangzhou Institute of Medicine (HIM), Chinese Academy of Sciences, Hangzhou, Zhejiang, China
[2] Key Laboratory of Prevention, Diagnosis and Therapy of Upper Gastrointestinal Cancer of Zhejiang Province, Hangzhou, Zhejiang, China

## ABSTRACT

**Background.** Pancreatic cancer is a highly aggressive and lethal disease with limited treatment options. In this study, we investigated the potential therapeutic effects of compound KL-6 on pancreatic cancer cells.

**Methods.** The study involved assessing the inhibitory effects of KL-6 on cell proliferation, clonogenic potential, cell cycle progression, apoptosis, migration, and invasion. Additionally, we examined the action mechanism of KL-6 by RNA-seq and bioinformatic analysis and validated by qRT-PCR and western blot in pancreatic cancer cells.

**Results.** Our results demonstrated that KL-6 effectively inhibited the growth of pancreatic cancer cells in a dose-dependent manner. It induced G2/M phase cell cycle arrest and apoptosis, disrupting the cell cycle progression and promoting cell death. KL-6 also exhibited inhibitory effects on cell migration and invasion, suggesting its potential to suppress the metastatic properties of pancreatic cancer cells. Furthermore, KL-6 modulated the expression of genes involved in various cancer-related pathways including apoptosis and ferroptosis.

**Conclusion.** These findings collectively support the potential of KL-6 as a promising therapeutic option for pancreatic cancer treatment. Further research is needed to fully understand the underlying mechanisms and evaluate the clinical efficacy of KL-6 in pancreatic cancer patients.

## INTRODUCTION

Pancreatic cancer is a highly aggressive and lethal malignancy with limited treatment options and a grim prognosis (*Wood et al., 2022*). It currently ranks as the fourth leading cause of cancer-related deaths worldwide, and its incidence continues to rise (*Rahib et al., 2021*). Often referred to as a "silent killer," pancreatic cancer is notorious for its lack of symptoms during its early stages, leading to late-stage diagnosis when the disease has already spread to distant organs (*Rawla, Sunkara & Gaduputi, 2019*). As a result, the prognosis for pancreatic cancer is generally poor, and survival rates remain low.

Corresponding authors
Jinyun Dong,
dongjinyun1989@163.com
Jieer Ying, jieerying@aliyun.com

The primary treatment approach for pancreatic cancer is surgical resection, which offers the best chance of a cure for patients with localized disease. However, only a small fraction of patients, approximately 20%, are suitable candidates for surgery at the time of diagnosis (*Manrai et al., 2021*). For the majority of patients, the treatment options are limited to chemotherapy and radiation therapy, which provide modest benefits in terms of prolonging survival and enhancing the quality of life. Unfortunately, the overall efficacy of chemotherapy and radiation therapy in improving survival rates remains limited (*Versteijne et al., 2022*; *Reyngold et al., 2021*; *Heinrich & Lang, 2017*). Consequently, there is an urgent need for the development of innovative therapeutic strategies and novel drugs to address the formidable challenges posed by pancreatic cancer.

In our previous study, we identified a benzochalcone derivative called KL-6 {(E)-1–(1-Hydroxy-4,5,8-trimethoxynaphthalen-2-yl)-3-(quinolin-6-yl) prop-2-en-1-one} that exhibited potent anti-cancer activity against gastric cancer cells (*Dong et al., 2022*). *In vitro* experiments showed that KL-6 inhibited the proliferation and invasion capabilities of gastric cancer cells and induced apoptosis. *In vivo,* experiments demonstrated that KL-6 suppressed tumor growth and metastasis at lower doses without causing significant toxic side effects such as weight loss or organ damage in mice. Additionally, STAT3 has been demonstrated to be prominently expressed in various cancer cells, including gastric cancer. Further mechanistic study revealed that KL-6 could concentration-dependently suppress STAT3 phosphorylation, which may partly contribute to its anticancer activity. The molecular modeling experiment revealed that hydrogen bond was a predominant factor for KL-6 tightly binding to STAT3 (*Dong et al., 2022*). This indicates that KL-6 has good safety characteristics, providing a basis for further clinical development. However, it remains unclear whether KL-6 possesses anti-tumor activity in pancreatic cancer and its underlying mechanism of action.

In recent years, because tumor cells show avoidance of apoptosis, which causes treatment resistance and recurrence, numerous studies have been devoted to alternative cancer cell mortality processes, namely necroptosis, pyroptosis, ferroptosis, and cuproptosis (*Tong et al., 2022*). Among them, the regulated cell death pathway, represented by iron death, has gained increasing importance as a target for cancer drug development (*Mukherjee et al., 2023*; *Zhang et al., 2022a*). Ferroptosis inducers have developed as anticancer drugs in preclinical and clinical settings for curing difficult-to-treat cancers, but pancreatic cancer, which is highly resistant to chemotherapy drugs and has the worst survival rate of all cancers. Therefore, we are committed to hopefully finding some iron death inducers as a way to increase hope for the treatment of pancreatic cancer. However, the mechanism of KL-6 inducing ferroptosis and activating antitumor immunity remains obscure.

Our study focuses on investigating the anti-pancreatic cancer activity of compound KL-6 and its potential mechanisms. The results indicate that KL-6 can activate both ferroptosis and apoptosis, which are important processes in cancer treatment. Additionally, the study suggests that the inhibition of PLAC8 expression may contribute to the induction of apoptosis by KL-6. These findings provide valuable insights into potential novel strategies for comprehensive treatment of pancreatic cancer.

## MATERIALS & METHODS

### Cell culture and compound

PANC-1 and Mia-PACA2 pancreatic cancer cells were obtained from the American Type Culture Collection (ATCC) and cultured in DMEM complete medium supplemented with 10% fetal bovine serum (FBS, GIBCO) and 100 U/mL penicillin-streptomycin solution (Biosharp, Hefei, China). The cells were incubated at 37 °C in a humidified atmosphere with 5% $CO_2$. KL-6 was prepared as previously described, starting materials were obtained from commercial suppliers and all solvents were used as supplied. Monitoring using proton nuclear magnetic resonance (1H NMR), 13 C NMR, thin-layer chromatography and column chromatography et al. (*Dong et al., 2022*).

### Cell viability assay

Approximately $5 \times 10^3$ PANC-1 and Mia-PACA2 cells were seeded in separate 96-well plates. After overnight adhesion, various concentrations (ranging from 0.1 to 100 μM) of test compounds and positive drugs were added for a 24-hour and 72-hour treatment period. Subsequently, 10 μL of CCK8 reagent (Biosharp, Hefei, China) was added to each well, and the absorbance at 450 nm was measured using a microplate reader (Tecan, Switzerland). Cell survival rates were calculated by comparing the results with those of the control group, enabling the evaluation of the effects of the tested compounds on the proliferation of different pancreatic cancer cells (*Guan et al., 2023*).

### Colony formation assay

In the colony formation assay, Mia-PACA2 pancreatic cancer cells were seeded in 6-well plates at a density of approximately $1 \times 10^3$ cells per well. The cells were then treated with either DMSO (control) or different concentrations (1 or 2 μM) of compound KL-6 for 24 h. After the treatment, the medium was replaced with fresh medium and the cells were allowed to grow for a period of 13 days. At the end of the incubation period, the cells were washed twice with PBS to remove any unattached cells. Then, the cells were fixed with 4% paraformaldehyde for 20 min to immobilize them. After fixation, the cells were stained with crystal violet for 20 min to visualize and quantify the colonies formed by the cells (*Zhang et al., 2020*). The stained cells were then photographed using a scanner. To count the colonies, image analysis software like ImageJ was used. The software can automatically detect and count the colonies based on their size, shape, and color.

### Cell cycle arrest assay

PANC-1 cells were seeded in a 10 cm cell culture dish at a density of $2 \times 10^6$ cells per dish and incubated for 24 h. The culture medium was then replaced with fresh medium containing different concentrations of compound KL-6 (0, 0.5, 1, 2 μM). After incubating for 24 h, the cells were trypsinized, collected, and fixed with ice-cold 95% ethanol overnight at 4 °C. Following fixation, the cells were washed with PBS and incubated with propidium iodide (PI) staining solution containing RNase A (BD Biosciences, San Jose, CA, USA) for 30 min at 37 °C in the dark. The cell cycle distribution was determined using a flow cytometer, and the data were analyzed using FlowJo software. The percentage of cells in

the G0/G1, S, and G2/M phases was quantified to assess the cell cycle arrest induced by compound KL-6 (*Zhang et al., 2019*).

## Cell apoptosis assay

To investigate the effect of compound KL-6 on apoptosis in pancreatic cancer cells, flow cytometry was employed. Firstly, PANC-1 cells were seeded in a 10 cm cell culture dish at a density of $2 \times 10^6$ cells per dish. Subsequently, the cells were treated with different groups: control group (DMSO) and KL-6 treatment groups at various concentrations (0.5, 1, 2 µM). The treatment duration was 24 h. After the treatment, the cells were collected and subjected to apoptosis analysis. The cells were first harvested by centrifugation and washed with PBS buffer. Then, the cells were performed according to the manufacturer's instructions (Elabscience, Wuhan, China) and re-suspended with 500 µL 1× binding buffer. Afterward, cell suspension was re-suspended in 5 µL fluorescein isothiocyanate-conjugated annexin plus 5 µL propidium iodide, followed by 15 min incubation at room temperature in the dark. Finally, flow cytometry was performed for apoptosis analysis. By setting appropriate wavelengths and detectors, the PI fluorescence signal in the cells was measured. By analyzing the DNA content in the cells, the apoptotic status of the cells could be determined. Cells with normal fluorescence were considered as viable cells, while cells with higher fluorescence were considered as apoptotic cells (*Zhang et al., 2019*).

## Wound healing assay

PANC-1 and Mia-PACA2 cells were seeded in 6-well plates and cultured until they reached 80–90% confluence in the wound healing assays. The culture medium was then replaced with 1% serum medium. The cells were treated with different concentrations of compound KL-6 (0, 1, 2 µM). A straight scratch was made using a pipette tip across the cell monolayer. The cells were then observed and photographed under a microscope at 0, 12, 24, and 36 h after the scratch was made to assess the migration of the cells at the active edge of the scratch (*Zhou et al., 2021*).

## Transwell migration assay

In the transwell migration assay, spread matrigel was added to the upper compartment of a transwell chamber and incubated for 6 h at 37 °C to form a gel layer. Then, $1 \times 10^4$ PANC-1 and Mia-PACA2 cells suspended in 200 µL of serum-free medium were plated in the upper chamber of the transwell. In the lower chamber, 600 µL of medium containing 10% fetal bovine serum and different concentrations of compound KL-6 (0, 1, 2, 5 µM) were added. The presence of serum and KL-6 in the lower chamber acts as a chemoattractant for the cells, stimulating their migration towards the lower chamber. The transwell chamber was then incubated for 48 h to allow the cells to migrate through the gel layer and toward the lower chamber. After the incubation period, the cells on the upper side of the gel layer were removed by swabbing with cotton swabs. The cells that migrated to the lower side of the gel layer and onto the bottom of the transwell chamber were fixed with 4% paraformaldehyde, washed three times with PBS, and stained with 2.5% crystal violet. The stained cells were then visualized and photographed using an inverted microscope. The cells that have migrated to the lower side of the gel layer appear as stained spots on the

bottom of the chamber. To quantify the migration, manual counting of the stained spots can be performed. The number of migrated cells can be determined by counting the stained spots in multiple fields of view and averaging the counts (*Zhou et al., 2021*).

## Intracellular LPO measurement

The level of lipid peroxidation (LPO) was measured using the C11-BODIPY 581/591 dye through flow cytometry (FCM). Specifically, PANC-1 cells were incubated in 6-well plates at a density of $5 \times 10^4$ cells per well for 24 h. Subsequently, the culture medium was replaced with a fresh medium containing different concentrations of compound KL-6 (0, 0.5, 1, and 2 $\mu$M). After incubating for 6 h, the cells were stained with the C11-BODIPY 581/591 probe (Invitrogen) for 20 min. Finally, the cells were harvested and analyzed using FCM (*Zhou et al., 2021*).

## RNA-sequencing and bioinformatics analysis

PANC-1 and Mia-PACA2 cells were seeded in 6-well plates at a density of $5 \times 10^4$ cells per well and incubated for 24 h. The culture medium was then replaced with fresh medium containing different concentrations of compound KL-6 (0 or 2 $\mu$M). After incubating for 24 h, the total RNA was extracted from the cells using a commercially available RNA extraction kit. The quality and quantity of the RNA samples were assessed using a NanoDrop spectrophotometer and an Agilent Bioanalyzer. Subsequently, RNA-sequencing libraries were prepared using a TruSeq RNA Library Preparation Kit according to the manufacturer's instructions. The libraries were then sequenced on an Illumina sequencing platform. Raw sequencing data were processed to obtain clean reads by removing low-quality reads, adapter sequences, and reads with ambiguous bases. The clean reads were aligned to the reference genome using a suitable alignment tool. The statistical power of this experimental design with a biological replication of 3, sequencing depth of 15 and coefficient of variation of 0.1, calculated in RNASeqPower is 0.87.

Gene expression levels were quantified and normalized using methods such as fragments per kilobase of transcript per million mapped reads (FPKM). Differential gene expression analysis was performed to identify genes that were significantly regulated by compound KL-6. Genes with absolute of log2(fold change) $\geq$ 0.58 and a false discovery rate (FDR) < 0.05 were considered differentially expressed. Kyoto Encyclopedia of Genes and Genomes (KEGG) pathway enrichment analyses were conducted to explore the biological functions and pathways associated with the differentially expressed genes (*Qi et al., 2022*).

## RNA extraction, cDNA synthesis, and quantitative real-time PCR

Total RNA was extracted from the PANC-1 and Mia-PACA2 cells using a commercially available RNA extraction kit (YiShan Biotechnology Co. Ltd., Shanghai, China) according to the manufacturer's instructions. The quality and quantity of the RNA samples were assessed using a NanoDrop spectrophotometer. For complementary DNA (cDNA) synthesis, 1 $\mu$g of total RNA was reverse transcribed into cDNA using a Fast-All-in-One RT Kit (YiShan Biotechnology Co. Ltd., Shanghai, China). The cDNA synthesis reaction was performed at 37 °C for 60 min, followed by heat inactivation at 95 °C for 5 min. The synthesized cDNA

samples were then subjected to quantitative real-time polymerase chain reaction (qRT-PCR) using specific primers corresponding to the target genes. The qRT-PCR reactions were performed by 2xSuper SYBR Green Qpcr Master Mix (YiShan Biotechnology Co. Ltd., Shanghai, China) on CFX96™ PCR instruments (Bio-Rad Laboratories, Hercules, CA, USA). The amplification conditions comprised an initial denaturation step at 95 °C for 10 min, followed by 40 cycles of denaturation at 95 °C for 15 s, annealing at 60 °C for 30 s, and extension at 72 °C for 30 s. Relative mRNA expression levels were calculated using the $2^{-\Delta\Delta Ct}$ method, with GAPDH as the reference gene for normalization (*Qi et al., 2022*). The experiment was performed in triplicate, and the results were expressed as the mean ± standard deviation.

The forward and reverse primer sequences for the genes were as follows:

PLAC8, Forward: 5′-GGGTGTCAAGTTGCAGCTGAT-3′and Reverse: 5′-TAGATCCAGGGATGCCATATCG-3′;

NME1-NME2, Forward: 5′-CTAAGCAGCTGGAAGGAACC-3′and Reverse: 5′-TAGTGCTGCTTCAGGTGTTC-3′;

SLC7A11, Forward: 5′-AGAGTAAGAACAATGGCTTCAGGAG-3′and Reverse: 5′-AGGTCAGACAGGCAGGTATCATC-3′;

GPX4, Forward: 5′-TAGAAATAGTGGGGCAGGTCC-3′and Reverse: 5′- CGTCAAATTCGATATGTTCAGC-3′;

## Western blotting

First, the cells of interest were treated with different concentrations of compound KL-6 for a specified period. After the treatment, the cells were lysed using a lysis buffer containing protease inhibitors to extract total protein. The protein concentration was determined using a BCA protein assay kit (Beyotime, Jiangsu, China). Equal amounts of protein samples were separated by SDS-PAGE gel electrophoresis and then transferred onto a PVDF membrane (Roche, Basel, Switzerland). The membrane was blocked with 5% non-fat milk in Tris-buffered saline with Tween 20 for 1 h at room temperature to prevent non-specific binding. Next, the membrane was incubated with the primary antibody (diluted according to the manufacturer's instructions) overnight at 4 °C, followed by washing with TBST. Then, the membrane was incubated with a secondary antibody conjugated with horseradish peroxidase for 1 h at room temperature. After washing, the protein bands were visualized using an enhanced chemiluminescence substrate and captured using an imaging system (*Guan et al., 2023*). Catalog numbers of the antibodies used in this experiment were as follows: DDX27 (1:1000 dilution, #2251C2a; Santa Cruz Biotechnology, Dallas, TX, USA), Bcl-2 (1:1000 dilution, #15071S; Cell Signaling Technology, Danvers, MA, USA), MCL1 (1:1000 dilution, #9762T; Cell Signaling Technology), Caspase 3 (1:1000 dilution, #9662S; Cell Signaling Technology), SLC7A11 (1:1000 dilution, #12691; Cell Signaling Technology), GPX4 (1:1000 dilution, #59735; Cell Signaling Technology), $\beta$-Actin (1:1000 dilution, #AF0003; Beyotime).

## Statistical analysis

Statistical analysis was conducted using GraphPad Prism version 8.0 software. Data were presented as the mean ± standard error of the mean (SEM). To compare survival curves,

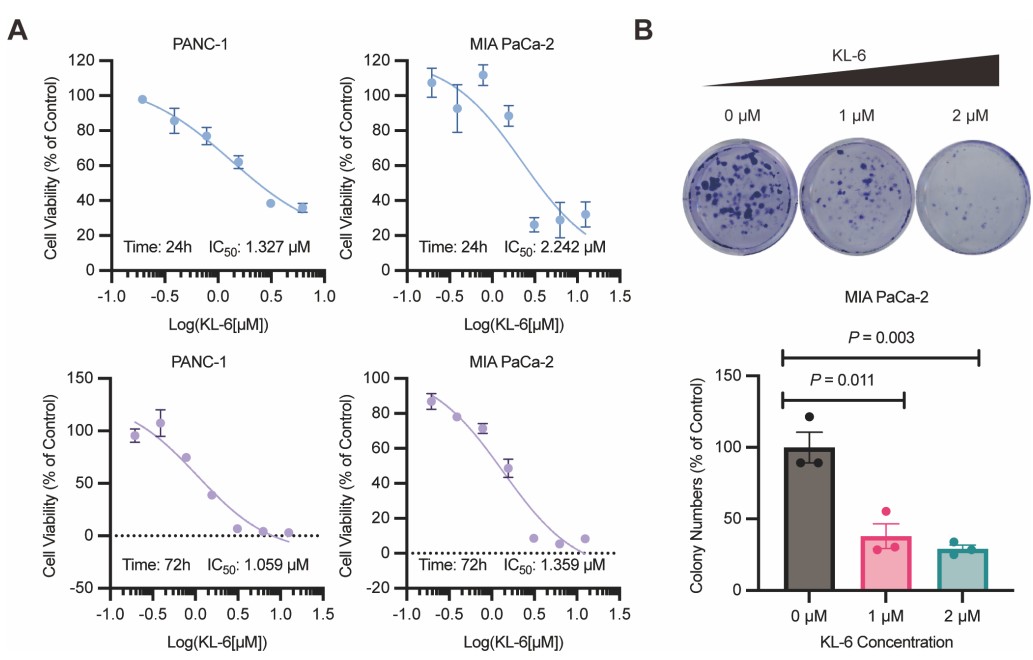

**Figure 1** **KL-6-mediated inhibition of cell proliferation and clonogenic potential.** (A) Viability of PANC-1 and Mia-PACA2 cells treated with KL-6 evaluated by 24-h and 72-h CCK8 assays. (B) Clonogenicity of Mia-PACA2 cells treated with different concentrations of compound KL-6.

the log-rank test was utilized. Statistical significance was considered at a *p*-value of less than 0.05 ($P < 0.05$).

## RESULTS

### KL-6 demonstrates potent inhibition of pancreatic cancer cell proliferation and clonogenic potential

The results of the 24-hour and 72-hour CCK-8 assays demonstrated that compound KL-6 inhibited the growth of pancreatic cancer cells PANC-1 and Mia-PACA2 (Fig. 1A). The $IC_{50}$ values at 24 h were 1.327 µM and 2.242 µM for PANC-1 and Mia-PACA2, respectively. At 72 h, the $IC_{50}$ values were 1.059 µM and 1.359 µM for PANC-1 and Mia-PACA2, respectively. These findings indicate that KL-6 has a dose-dependent inhibitory effect on the proliferation of these pancreatic cancer cell lines.

Furthermore, the colony formation assay showed that KL-6 significantly suppressed the proliferation and clonogenic potential of Mia-PACA2 cells (Fig. 1B). At a concentration of 1 µM KL-6, the colony formation ability of Mia-PACA2 cells was inhibited by 61.9%. At a concentration of 2 µM KL-6, the colony formation ability was further inhibited, with 70.8% inhibition in Mia-PACA2 cells. These results suggest that KL-6 has a potent inhibitory effect on the clonogenic potential of pancreatic cancer cells, indicating its potential as a therapeutic agent in pancreatic cancer treatment.

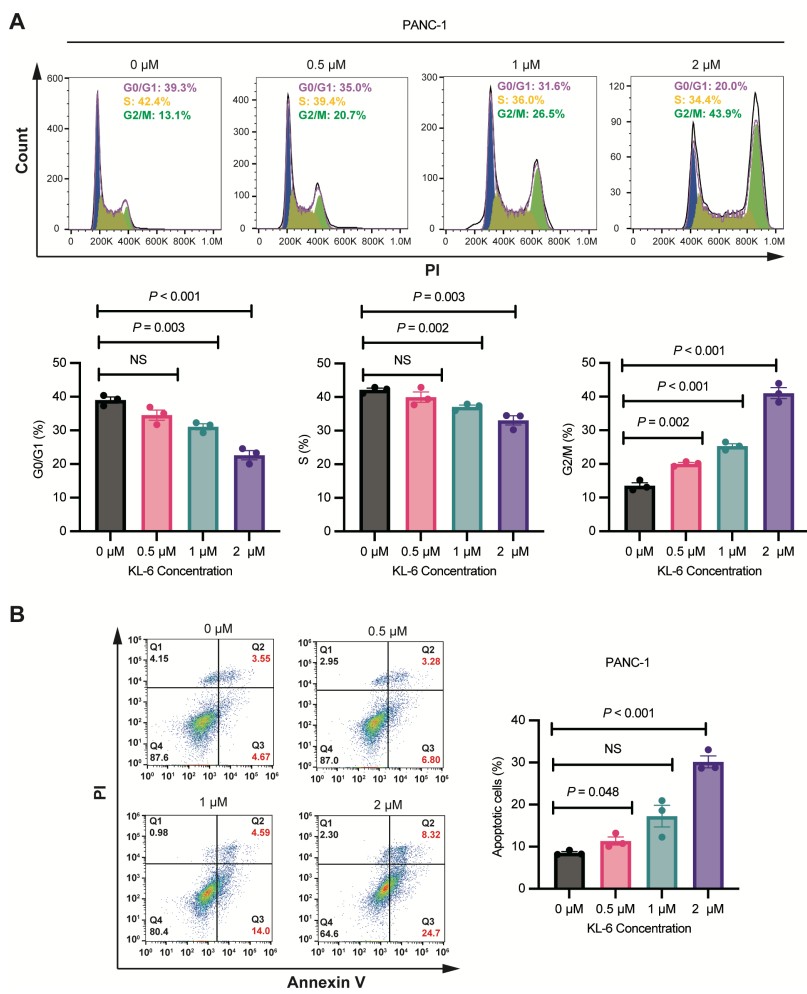

**Figure 2  KL-6-induced cell cycle arrest in G2/M phase and apoptosis in PANC-1 cells.** (A) Analysis of cell cycle distribution in PANC-1 cells treated with KL-6 for 24 h using propidium iodide/RNase. (B) Detection of apoptosis in Mia-PACA2 cells treated with compound KL-6 at varying concentrations for 48 h using FITC-Annexin V assay.

## Concentration-dependent induction of G2/M phase cell cycle arrest and apoptosis by KL-6 in PANC-1 cells

As shown in Fig. 2A, Cell cycle analysis revealed that treatment with different concentrations of KL-6 (0, 0.5, 1, 2 μM) resulted in a concentration-dependent induction of cell cycle arrest in the G2/M phase in PANC-1 cells. At 2 μM KL-6 concentration, the percentage of cells arrested in the G2/M phase reached 41.03%. Additionally, the percentage of cells in the G0/G1 phase decreased with increasing KL-6 concentration, with a significant reduction observed at 1 μM and 2 μM. Similarly, the percentage of cells in the S phase also decreased with increasing KL-6 concentration, reaching a significant reduction at 1 μM and 2 μM. These findings demonstrate that KL-6 treatment leads to cell cycle arrest in the G2/M phase and decreases the proportion of cells in the G0/G1 and S phases in PANC-1 cells.

Figure 2B showed that KL-6 treatment led to a dose-dependent increase in the percentage of both early and late apoptotic cells in PANC-1 cells. Compared to the control group, the percentage of total apoptotic cells significantly increased at 0.5 μM and 2 μM KL-6 concentrations, with the respective increases 2.82%, and 21.64%. These results indicate that KL-6 induces apoptosis in PANC-1 cells in a concentration-dependent manner.

## KL-6 inhibits the pancreatic cells' migration and invasion

Wound healing assay demonstrated that different concentrations of KL-6 inhibited cell migration ability. At 24 h, 1 μM KL-6 significantly inhibited the migration of 16.95% PANC-1 and 21.71% Mia-PACA2 cells, while 2 μM KL-6 inhibited the migration of 12.77% PANC-1 and 20.22% Mia-PACA2 cells (Fig. 3A). Transwell assay showed that KL-6 concentration-dependently inhibited the invasion of PANC-1 and Mia-PACA2 cells (0, 1, 2, 5 μM). At 2 and 5 μM KL-6, the invasion of 78.75% and 75.83% PANC-1 cells was significantly inhibited. Furthermore, at 1, 2 and 5 μM KL-6, the invasion of 21.71%, 57.31% and 87.16% Mia-PACA2 cells was significantly inhibited, respectively (Fig. 3B). These findings suggest that KL-6 inhibits cell migration and invasion in a concentration-dependent manner in PANC-1 and Mia-PACA2 cells.

## KL-6 treatment resulted in the differential regulation of gene expression in pancreatic cancer cells, involving various cancer-related pathways

Our study yielded significant findings on the effects of KL-6 treatment on gene expression in PANC-1 and Mia-PACA2 cells. We observed that KL-6 caused the downregulation of 401 and 102 genes, and the upregulation of 574 and 147 genes, respectively, in PANC-1 and Mia-PACA2 cells as shown in Fig. 4B. KEGG pathway analysis revealed that KL-6 influenced the p53 signaling pathway, which is consistent with our previous studies. Additionally, KL-6 was found to modulate a range of pathways implicated in cancer progression, including cell cycle, ferroptosis, apoptosis, and epithelial-mesenchymal transition (EMT)-associated pathways such as focal adhesion and tight junction as illustrated in Fig. 4A. To further explore KL-6′s mode of action in pancreatic cancer treatment, we identified the differential expression of genes that were shared between both cell lines. Based on the results, the five genes that are downregulated in both PANC-1 and Mia-PACA2 cells are NME1-NME2, PAGE1, ASB9, TMEM14A, and PLAC8. Additionally, 13 genes are upregulated, including RHOT2, KLHL17, JUN, SMTN, TMEFF1, PKD1P6, SPHK2, NCOR2, SHANK3, MIF-AS1, LIF, and ANKLE1 (Fig. 4B). These genes have diverse functions and their dysregulation in cancer cells may contribute to tumor development, progression, and response to treatment (*Salomaa et al., 2021*; *Li et al., 2018*; *Shi et al., 2019*; *Jiang et al., 2023*). Interestingly, TCGA and GTEx data demonstrated higher expression levels of PLAC8, NME1-NME2, and TMEM14A in pancreatic cancer patients as illustrated in Fig. 5A. The Kaplan–Meier survival curves and log-rank test showed that high expression levels of PLAC8 and NME1-NME2 were significantly associated with poor overall survival in pancreatic carcinoma, as demonstrated in Fig. 5B.

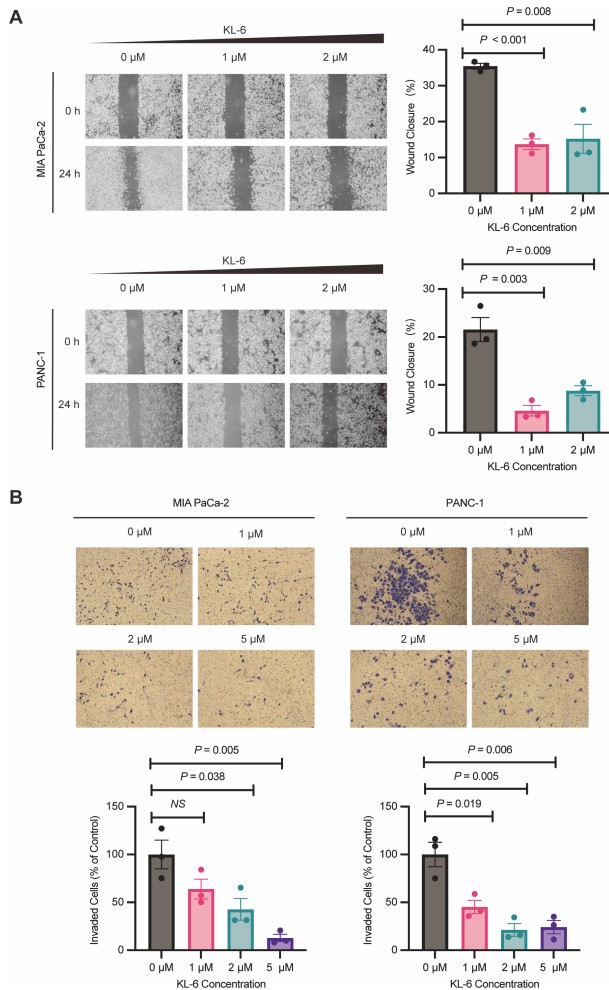

**Figure 3** **KL-6 inhibits the migration and invasion of pancreatic cells.** (A) Representative photomicrographs showing PANC-1 and Mia PACA2cells treated with compound KL-6 at indicated concentrations for 24 h using the wound healing assay. (B) Transwell assay of PANC-1 and Mia PACA2 cells after 48 h of treatment with compound KL-6 at indicated concentrations.

## KL-6 treatment caused changes in lipid peroxidation, and induced apoptosis in PANC-1 and Mia-PACA2 cells

To validate our gene expression profiling, we performed qRT-PCR to measure PLAC8 and NME1-NME2 levels in PANC-1 cells treated with variable concentrations of KL-6. The results indicated that KL-6 treatment led to a reduction in PLAC8 mRNA levels (Fig. 6A). PLAC8, also known as onzin, was found to be strongly expressed in advanced preoplastic lesions and invasive human pancreatic ductal adenocarcinoma (PDAC). Recent evidence suggests that in pancreatic cancer cells, PLAC8 localizes to the inner face of the plasma membrane and interacts with specific membranous structures in a stable manner (*Zhou et al., 2021*). Inhibiting PLAC8 expression significantly inhibits pancreatic cancer cell growth by affecting cell-cycle progression and modifying key cell-cycle regulators (*Zhou et al., 2021*). Additionally, deficiency of PLAC8 in mouse models of pancreatic cancer

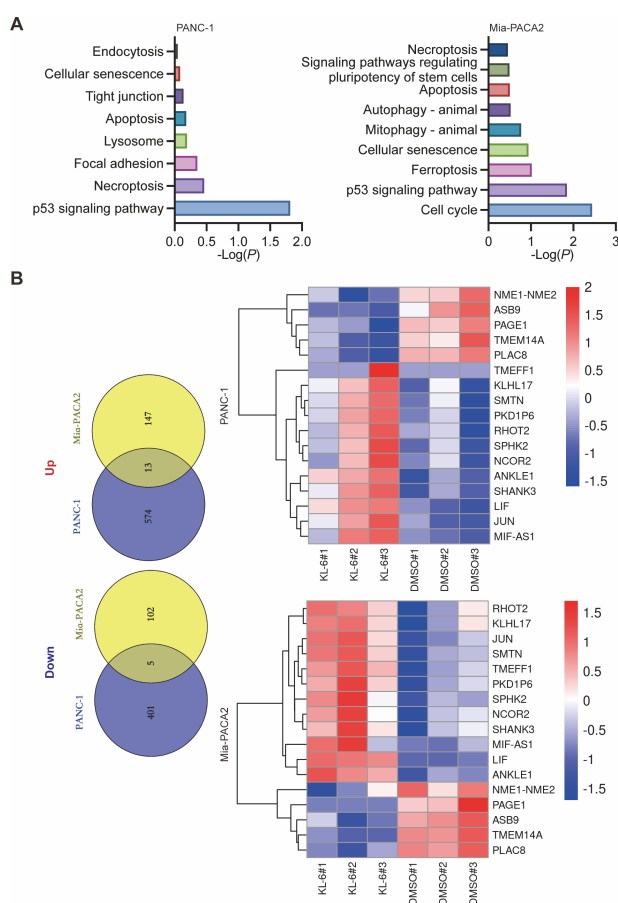

**Figure 4** **Dysregulated pathways identified in PANC-1 and Mia-PACA2 cells through transcriptome analysis.** (A) KEGG analysis for the dysregulated genes by KL-6 compared to control. (B) Venn plot and heatmap displaying the differentially expressed genes between KL-6 treated and untreated cells.

inhibits tumor formation (*Kaistha et al., 2016*). These findings establish PLAC8 as a central mediator of tumor progression in PDAC and a potential target for diagnosis and therapy. In contrast, NME1-NME2 levels remained unaffected. Moreover, in PANC-1 and Mia-PACA2 cells, KL-6 treatment reduced Bcl-2 and MCL1, two genes that suppress apoptosis, indicating that KL-6 also triggers apoptosis (Fig. 6D). Previous studies showed that DDX27 regulated colony formation *via* cell cycle control and was a predictor of poor prognosis in gastric and colorectal cancer (*Bear, Vonderheide & O'Hara, 2020*). Our results indicated that KL-6 treatment decreased DDX27 expression in both PANC-1 and Mia-PACA2 cells (Fig. 6D).

Our transcriptomic analysis raised questions about KL-6's role in ferroptosis, leading us to explore lipid peroxidation as a potential marker of ROS. We employed the BODIPY 581/591C11 probe for lipid peroxidation detection and utilized flow cytometry for analysis. Our findings demonstrated that increasing KL-6 concentrations led to a visible dose-dependent progression in lipid peroxidation levels when compared to the control group (Fig. 6B). In the process of ferroptosis, GPX4 utilizes reduced GSH to convert phospholipid

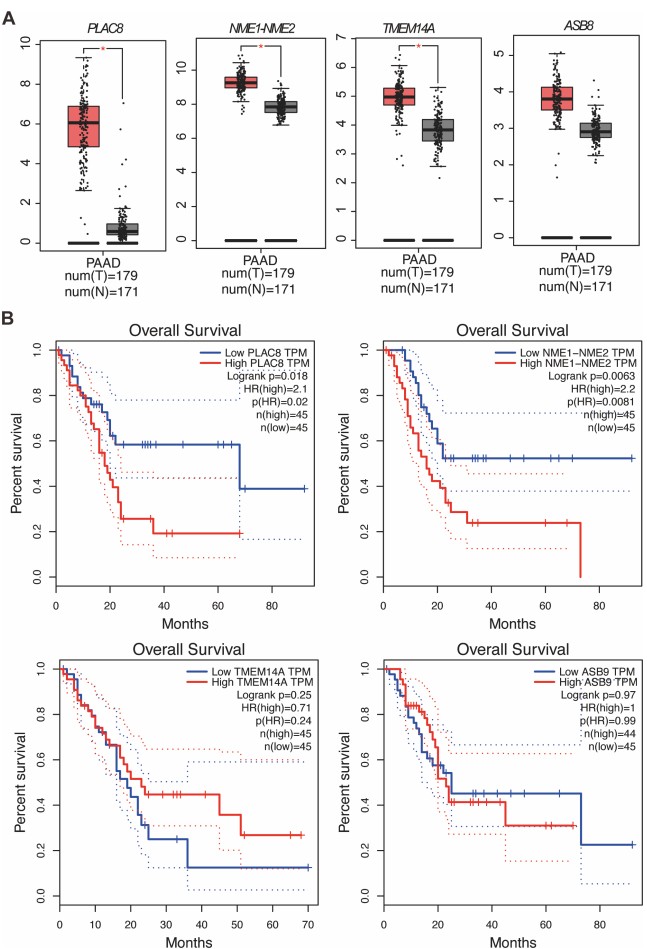

**Figure 5   MRNA levels and effects on overall survival rates of indicated genes.** (A) The mRNA levels of indicated genes in pancreatic cancer tissues compared to normal tissues in the GEPIA dataset. The mean and standard deviation (SD) are shown. (B) Kaplan–Meier analysis of the overall survival rate of pancreatic cancer patients based on the mRNA levels of the indicated genes.

hydroperoxides to lipid alcohols and inhibits ferroptosis (*Miao et al., 2022*). In addition, recent studies revealed that overexpression of SLC7A11, a protein that imported cysteine for glutathione biosynthesis and antioxidant defense, promotes tumor growth partly through suppressing ferroptosis (*Koppula, Zhuang & Gan, 2021*). To clarify these changes, we used qRT-PCR and western blot to evaluate SLC7A11 and GPX4 levels, two well-known genes linked to ferroptosis. Our results indicated that KL-6 treatment decreased SLC7A11 mRNA expression in PANC-1 cells, but not GPX4 expression (Fig. 6C). Nonetheless, SLC7A11 and GPX4 protein expression did not alter with KL-6 treatment (Fig. 6D), suggesting that KL-6 may affect ferroptosis *via* other pathways.

## DISCUSSION

Additionally, the KRAS oncogene plays a critical role in the initiation and maintenance of pancreatic tumors and its signaling network represents a major target for therapeutic

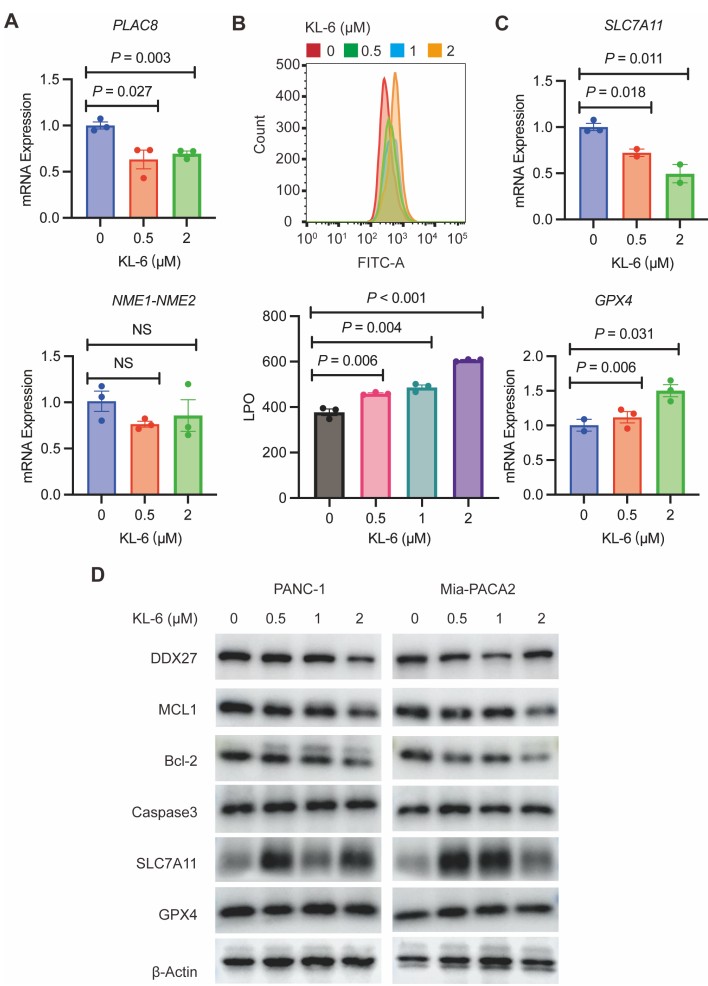

**Figure 6 KL-6-induced increase in lipid peroxidation levels and apoptosis *via* inhibition of PLAC8.**
(A) The mRNA expression levels of PLAC8 and NME1-NME2 in PANC-1 cells after treatment with different concentrations of KL-6. (B) The detection of lipid peroxidation levels using BODIPY 581/591C11. (C) The mRNA expression levels of SLC7A11 and GPX4 in PANC-1 cells treated with varying concentrations of KL-6. (D) Western blot analysis results showing the impacts of KL-6 on DDX27, MCL1, Bcl-2, Caspase3, SLC7A11, and GPX4 in PANC-1 and Mia-PACA2 cells.

intervention. Several inhibitors have been developed against kinase effectors in various Ras signaling pathways (*Luo, 2021*). The first approved, EGFR inhibitor erlotinib in combination with gemcitabine, gave a modest survival benefit compared to gemcitabine alone and was effectively abandoned by the community after negative data in KRAS-mutant colorectal cancer emerged (*Luo, 2021*; *Moore et al., 2007*). More recently, the TRK kinase inhibitors larotrectinib and entrectinib have been approved for solid tumors harboring a NTRK-fusion gene, but NTRK-fusion occurs in only 0.5% of pancreatic ductal adenocarcinoma (PDAC) and the benefit of TRK inhibitors have not been systematically evaluated in PDAC beyond individual cases (*Luo, 2021*; *Drilon et al., 2018*). Here, we identified a benzochalcone derivative called KL-6 that exhibits with promising

anticancer efficacy in pancreatic cancer cells. KL-6 effectively inhibits pancreatic cancer cell proliferation and clonogenic potential in a dose-dependent manner. It induces G2/M phase cell cycle arrest and apoptosis, disrupting the cell cycle progression and promoting cell death. KL-6 also inhibits cell migration and invasion, suggesting its ability to suppress the metastatic properties of pancreatic cancer cells. Furthermore, KL-6 modulates the expression of genes involved in various cancer-related pathways, highlighting its multi-faceted mechanism of action.

In recent years, there has been a growing interest in targeted therapies for pancreatic cancer (*Manrai et al., 2021*). These therapies aim to specifically target the molecular abnormalities and signaling pathways that drive the growth and spread of pancreatic cancer cells (*Piper et al., 2023*). For example, inhibitors of the epidermal growth factor receptor (EGFR) such as erlotinib have shown some efficacy in combination with chemotherapy (*Hammel et al., 2016*). Another promising approach is immunotherapy, which harnesses the body's immune system to recognize and attack cancer cells. Immune checkpoint inhibitors, such as pembrolizumab and nivolumab, have shown promise in early clinical trials for pancreatic cancer (*Bear, Vonderheide & O'Hara, 2020*). In addition to targeted therapies and immunotherapy, researchers are also exploring the use of novel drug delivery systems, such as nanoparticles and liposomes, to improve the delivery of chemotherapy drugs to the tumor site and enhance their effectiveness (*Gao et al., 2022*; *Bockorny et al., 2021*; *Yuan et al., 2022*). Overall, while the treatment landscape for pancreatic cancer remains challenging (*Halbrook et al., 2023*). Our findings collectively support the potential of KL-6 as a promising therapeutic option for pancreatic cancer treatment.

Next, a number of drugs induce ferroptosis in PDAC cells by targeting regulatory mechanisms, such as the antimalarial drug artesunate and the antiviral drug zalcitabine (*Zhang et al., 2022b*). In addition, other key molecules in the ferroptosis pathway, such as the intracellular levels of iron, SLC7A11, GSH, ROS, HSPA5, can be used to target ferroptosis for therapy (*Zhang et al., 2022b*; *Liang et al., 2019*). In vitro, we revealed that KL-6 reduced pancreatic cancer cell viability and clonogenesis. Next, we set out to explore the molecular mechanism by which KL-6 induces ferroptosis. At first, we found that KL-6 regulate SLC7A11 mRNA expression, but may not regulate SLC7A11 protein expression. In addition, we employed the BODIPY 581/591C11 probe for lipid peroxidation detection and utilized flow cytometry for analysis. Our findings demonstrated that increasing KL-6 concentrations led to a visible dose-dependent progression in lipid peroxidation levels when compared to the control group.

Additionally, KL-6 treatment leads to changes in lipid peroxidation levels and triggers apoptosis in pancreatic cancer cells. KL-6 treatment led to a reduction in PLAC8 mRNA levels. PLAC8 was initially discovered in screens for genes expressed in the placenta and was found associated with the spongiotrophoblast (*Mukherjee et al., 2023*; *Chen et al., 2022*). Recent studies have demonstrated that in addition to pancreatic cancer mentioned above, PLAC8 is also highly expressed in lung cancer tissue and serum (*Mukherjee et al., 2023*; *Chen et al., 2022*). Overexpression of PLAC8 stimulates cell proliferation and cancer development both *in vivo* and *in vitro* while silencing PLAC8 significantly suppresses

tumor growth (*Mukherjee et al., 2023*; *Chen et al., 2022*). Previous studies also showed that PLAC8 promotes lung cancer cell growth by activating the Wnt/ $\beta$-Catenin signaling pathway (*Chen et al., 2022*). Therefore, from a practical standpoint, we propose that PLAC8 holds promise as a potential treatment target for pancreatic cancer, providing a foundation and new insights for targeted therapy in pancreatic cancer patients. KL-6 treatment also resulted in the downregulation of Bcl-2 and MCL1, two genes that suppress apoptosis, suggesting that KL-6 triggers apoptosis in pancreatic cancer cells.

There are also some limitations of this study that need to be considered. First, we found that KL-6 regulates the expression of genes involved in various cancer-related pathways, including apoptosis and iron apoptosis; however, the exact mechanisms need to be further elucidated. Secondly, we did not perform an *in vivo* model for evaluation. Although KL-6 inhibited apoptosis and oxidative stress in pancreatic cancer cells, the interaction between apoptosis and oxidative stress was not clear, and the related signaling pathways need to be further confirmed. Finally, further investigation is needed to elucidate the exact mechanisms involved.

## CONCLUSION

Based on the results of our study, it can be concluded that compound KL-6 demonstrates significant potential as a therapeutic agent in the treatment of pancreatic cancer. Further research is warranted to fully elucidate the underlying mechanisms and evaluate the clinical efficacy of KL-6 in pancreatic cancer patients.

**Abbreviations**

| | |
|---|---|
| **ATCC** | American Type Culture Collection |
| **FBS** | fetal bovine serum |
| **CCK8** | cell counting kit 8 |
| **LPO** | lipid peroxidation |
| **FCM** | flow cytometry |
| **FPKM** | fragments per kilobase of transcript per million mapped reads |
| **FDR** | false discovery rate |
| **KEGG** | Kyoto Encyclopedia of Genes and Genomes |
| **cDNA** | complementary DNA |
| **qRT-PCR** | quantitative real-time polymerase chain reaction |

## ACKNOWLEDGEMENTS

We thank the Shared Instrumentation Core Facility and specialists' technical support at the Hangzhou Institute of Medicine (HIM), Chinese Academy of Sciences, for the use of instruments.

### Funding
This work was supported by the Zhejiang Provincial Natural Science Foundation of China (LQ22H280022). The funders had no role in study design, data collection and analysis, decision to publish, or preparation of the manuscript.

### Grant Disclosures
The following grant information was disclosed by the authors:
The Zhejiang Provincial Natural Science Foundation of China: LQ22H280022.

### Competing Interests
The authors declare there are no competing interests.

### Author Contributions
- Xiaoqing Guan conceived and designed the experiments, analyzed the data, prepared figures and/or tables, authored or reviewed drafts of the article, and approved the final draft.
- Bing Zhao performed the experiments, prepared figures and/or tables, and approved the final draft.
- Xiaodan Guan performed the experiments, prepared figures and/or tables, and approved the final draft.
- Jinyun Dong performed the experiments, authored or reviewed drafts of the article, and approved the final draft.
- Jieer Ying conceived and designed the experiments, authored or reviewed drafts of the article, and approved the final draft.

### DNA Deposition
The following information was supplied regarding the deposition of DNA sequences:
The RNA-seq data are available at the SRA: PRJNA1005062.

### Data Availability
The raw data are available in the Supplemental Files.

### Supplemental Information
Supplemental information for this article can be found online at http://dx.doi.org/10.7717/peerj.16291#supplemental-information.

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
