# Peer review of "A benzochalcone derivative synchronously induces apoptosis and ferroptosis in pancreatic cancer cells"

_PeerJ, doi:10.7717/peerj.16291_

## Round 0.1 · original submission · Major Revisions

Please address the reviewers' comments carefully.

Reviewer 1 ·

Basic reporting

In this manuscript, Guan and colleagues reported that a benzochalcone derivative, termed KL-6 suppresses the growth of pancreatic cancer cells by synchronously inducing apoptosis and ferroptosis. They also showed that KL-6 inhibited cell proliferation, caused cell cycle arrest at G2/M phase, prevented cell migration and invasion in pancreatic cancer cells. Moreover, they found that KL-6 treatment exhibited its anti-pancreatic cancer activity by modulating multiple signaling pathways and genes, which are highly expressed in pancreatic cancer and significantly correlated with patient survival. They also validated KL-6-mediated expression changes of genes at mRNA and protein levels.

Experimental design

The following concerns may be considered to improve the paper.

1. The authors should discuss more about the targeted therapies in both clinical and preclinical evaluation for pancreatic cancer patients, e.g., KRAS inhibitors and its efficacy and potential drug resistance in clinical studies.

2. Line 81. The authors should provide more details about the compound KL-6, e.g., purity, source of the compound, etc.

3. The authors should justify why they selected Panc1 and Mia-Paca-2 as the pancreatic cancer cell models in this study?

4. Line 286, between both “cell types” or “cell lines”?

5. The manuscript should be carefully proofread, and some minor issues should be addressed, e.g., IC50, 50 should be a subscript.

Validity of the findings

Overall, this study is interesting, and the manuscript is well-written.

Reviewer 2 ·

Basic reporting

The paper by Guan et al. is interesting and the results are promising. However, I still have some concerns about the current manuscript. Further modification and revision by addressing the concerns below may help improve the paper.

1. Because the authors demonstrated that KL-6 induced ferroptosis in pancreatic cancer cells. They should provide a brief introduction about the role of ferroptosis in pancreatic cancer development and progression. Why induction of ferroptosis should be noticed in this study? Is there any ferroptosis inducers developed as anticancer drugs in preclinical and clinical settings?

2. Did the authors upload the raw data of RNA-sequencing to a public database? It should be noticed that the efficacy of the compound KL-6 was promising and other researchers may be interested in this study and want to follow up it by analyzing the RNA-seq data.

3. The authors should provide more details about the use of antibodies in western blotting assays, e.g., the dilutions of each antibody should be provided.

4. Line 262, please doublecheck these numbers. Did the authors mean the percentages of apoptosis cells increased or increased to 2.82% and 21.64% after treated with 0.5 and 2 uM KL-6?

5. Figure 6D, the authors should provide the rationale for examining the expression levels of these proteins. Why the authors believe that they play important roles in the anticancer activity of KL-6?

6. The authors emphasized the role of induction of ferroptosis in the anti-pancreatic cancer activity of KL-6. However, they did not sufficiently discuss this effect. More importantly, the author should provide more discussion about the potential benefits of ferroptosis induction in the clinical treatment of pancreatic cancer patients.

7. The authors should discuss the limitations and future direction of this study in the Discussion section, including, but not limited to, the efficacy, toxicity, molecular targets, and mechanisms of action.

Experimental design

no comment

Validity of the findings

no comment

Additional comments

no comment

Reviewer 3 ·

Basic reporting

no comment

Experimental design

no comment

Validity of the findings

In the present study, Guan et al reported a potential anti-panceatic cancer drug KL-6 and demonstrated that the compound suppresses pancreatic cancer cell growth and metastasis by inhibiting cell proliferation, inducing cell cycle arrest, apoptosis, ferroptosis, and suppressing cell migration and invasion. Its potential molecular mechanisms were explored by analyzing the RNA-seq data and the inhibitory effects of KL-6 on the expression of several genes and proteins related to anti-apoptosis and anti-ferropotosis were showed. This manuscript can be published if the following questions can be addressed.


1. In the previous study, the author identified the compound KL-6 as a STAT3 inhibitor and demonstrated its anti-gastric cancer activity in vitro and in vivo. The authors should provide more details about this compound, especially its inhibitory effects on STAT3 signaling pathway.

2. Although the authors had provided details about the performance of these in vitro studies, they still need to add references for these methods and assays, including cell viability assay, colony formation assay, cell cycle arrest assay, cell apoptosis assay, wound healing assay, transwell migration assay, and intracellular LPO measurement.

3. Line 253, the percentage of cells arrested in the G2/M phase reached X%? Please carefully read and revise it.

4. Line 304-307, please provide references for these statements.

5. Lines 351-356, please provide references for the statements about PLAC8’s role in cancer.

Additional comments

no comment

---

## Round 0.2 · accepted · Accept

The authors have addressed the comments from the editor and reviewers.

Reviewer 1 ·

Basic reporting

This version addressed my concerns.

Experimental design

This version addressed my concerns.

Validity of the findings

This version addressed my concerns.

Additional comments

This version addressed my concerns.

Reviewer 2 ·

Basic reporting

Authors have addressed my concerns and I recommend acceptance.

Experimental design

no any concerns

Validity of the findings

no any concerns

Additional comments

no any concerns

Reviewer 3 ·

Basic reporting

/

Experimental design

/

Validity of the findings

The authors have answered all my questions.

Additional comments

/